# Altered Drop Jump Landing Biomechanics Following Eccentric Exercise-Induced Muscle Damage

**DOI:** 10.3390/sports9020024

**Published:** 2021-02-05

**Authors:** Themistoklis Tsatalas, Evangeli Karampina, Minas A. Mina, Dimitrios A. Patikas, Vasiliki C. Laschou, Aggelos Pappas, Athanasios Z. Jamurtas, Yiannis Koutedakis, Giannis Giakas

**Affiliations:** 1Department of Physical Education and Sport Science, University of Thessaly, 382 21 Trikala, Greece; evakapar@hotmail.gr (E.K.); lavassia123@gmail.com (V.C.L.); apappa66@yahoo.com (A.P.); ajamurt@pe.uth.gr (A.Z.J.); y.koutedakis@uth.gr (Y.K.); ggiakas@uth.gr (G.G.); 2Human Sciences Research Centre, College of Science and Engineering, University of Derby, Derby DE22 1GB 1, UK; M.Mina@derby.ac.uk; 3School of Physical Education and Sport Science at Serres, Faculty of Physical Education and Sport Sciences, Aristotle University of Thessaloniki, 541 24 Thessaloniki, Greece; dpatikas@auth.gr; 4Faculty of Arts, Wolverhampton University, Wolverhampton WV1 1LY, UK

**Keywords:** landing, isokinetic, exercise-induced muscle damage, fixture congestion

## Abstract

Limited research exists in the literature regarding the biomechanics of the jump-landing sequence in individuals that experience symptoms of muscle damage. The present study investigated the effects of knee localized muscle damage on sagittal plane landing biomechanics during drop vertical jump (DVJ). Thirteen regional level athletes performed five sets of 15 maximal eccentric voluntary contractions of the knee extensors of both legs at 60°/s. Pelvic and lower body kinematics and kinetics were measured pre- and 48 h post-eccentric exercise. The examination of muscle damage indicators included isometric torque, muscle soreness, and serum creatine kinase (CK) activity. The results revealed that all indicators changed significantly following eccentric exercise (*p* < 0.05). Peak knee and hip joint flexion as well as peak anterior pelvic tilt significantly increased, whereas vertical ground reaction force (GRF), internal knee extension moment, and knee joint stiffness significantly decreased during landing (*p* < 0.05). Therefore, the participants displayed a softer landing pattern following knee-localized eccentric exercise while being in a muscle-damaged state. This observation provides new insights on how the DVJ landing kinematics and kinetics alter to compensate the impaired function of the knee extensors following exercise-induced muscle damage (EIMD) and residual muscle soreness 48 h post-exercise.

## 1. Introduction

Jumping is an integral component of many sports (e.g., basketball, handball, and volleyball) and is accompanied by a landing due to the presence of gravitational acceleration. The athlete is required to change the motion of the center of mass (CoM) to stop the movement completely or to perform a subsequent task (e.g., jumping, change of direction). The jump-landing sequence is associated with an increased injury risk to the musculoskeletal structures that provide support to the knee joint, particularly to the anterior cruciate ligament (ACL). Up to 70–80% of ACL tears occur in non-contact situations (i.e., when there is no direct application of force from an opposing player) that include rapid deceleration, sudden change of direction, and landing [1]. ACL tears can be financially detrimental due to required reconstruction and rehabilitation and can affect the quality of life of otherwise healthy individuals [2]. In this regard, previous research has shown extensive interest on the effects of several factors such as type of sport, age, gender, fatigue, and previous injury [3,4,5,6,7] on specific biomechanical parameters while performing jump-landing activities, with a number of studies investigating their relation to potential injury risk [5,7,8,9,10]. 

One of the abovementioned factors that is considered an important sport-specific constraint is the development of acute (i.e., <3 h post-gameplay) and residual (72–96 h post-gameplay) fatigue [11]. The former is mainly caused by transient metabolic factors, whereas the latter is associated with the amount and density of intense eccentric muscular contractions during typical movement patterns (e.g., deceleration actions, multiple jump-landing sequences) seen in gameplay, and with the increase in physical demands given the short between-game recovery periods (i.e., fixture congestion) [9,11]. This is commonly known as exercise-induced muscle damage (EIMD) and is caused by strain to the muscle fibers during the aforementioned eccentric contractions that may result in disruption of the muscle cell structure [12]. Neuro-mechanical changes (e.g., force reduction), delayed onset of muscle soreness (DOMS), perturbations in biochemical parameters (e.g., creatine kinase), physical performance decrement (e.g., vertical jump ability), and decreased position sense are often reported for at least 3–5 days post-exercise [12,13]. Athletes often train or compete in games on consecutive days despite the presence of muscle damage. Previous research has reported an increased injury rate in players who participate in gameplay over short-congested periods and experience EIMD symptoms [14,15]. Therefore, inadequate time for rest and regeneration may not only limit performance and impair training quality but may also increase the risk of injury.

In this context, a comprehensive body of research motivated by a possible link between acute fatigue and landing injuries has examined the effects of fatigue on the loading of the musculoskeletal system during various landing tasks, although equivocal data exist in the literature [8,9]. However, the majority of EIMD studies have used different experimental protocols (e.g., isokinetic eccentric exercise, downhill running) to determine the effects on muscle damage indicators, such as peak torque during single-joint movements, DOMS, range of motion (ROM), and biochemical parameters [12,13]. Conversely, only a few publications refer to the effects of EIMD on mechanical loading during running, sidestep cutting, and activities of daily living (ADLs) [16,17,18,19,20,21,22,23,24], proposing that EIMD-specific biomechanical alterations may have injury implications [17,22]. Regarding jump-landing sequences, EIMD research has been limited to investigating performance (i.e., jump height) decrements [25,26,27] and neuromuscular alterations [26,27] in stretch-shortening cycle (SSC) activities, such as countermovement (CMJ) and drop vertical jump (DVJ). To our knowledge, the effects of EIMD, particularly of the knee muscles on the mechanical loading during landing, has not been explored in the literature. The stabilization of the knee joint is dependent on the absorption capability of the lower extremity musculature during landing activities. Elucidating the effects of EIMD on loading demands during landing could provide valuable knowledge to strength and conditioning practitioners to optimally manage training load (e.g., plyometric training volume and intensity) to restore athletes’ performance and reduce the potential risk of injury in subsequent competition. Therefore, the present study investigated the effects of EIMD on landing biomechanics. This was addressed by investigating the adaptations in sagittal plane kinematics and kinetics during the landing phase of a bilateral DVJ following the implementation of a single-joint EIMD protocol. It was hypothesized that the participants will display changes in landing mechanics due to DOMS and reduced force capacity whilst in a muscle-damaged state. 

## 2. Materials and Methods

### 2.1. Participants

Fifteen (*n* = 15) regional level athletes (i.e., basketball, football, and handball) with ≥5 year training experience in jump-landing activities volunteered to participate in the present study. The participants had no history of ACL injury and neurologic disorder or other lower extremity injuries within 12 months prior to participating in the study. Two participants did not manifest similar muscle damage indicators (strength and/or muscle soreness alteration) in both lower limbs and were thus excluded from the dataset, resulting in 13 (*n* = 13) total participants (age: 21 ± 2 years; body height: 181.1 ± 6.3 cm; body mass: 78.9 ± 8.5 kg). They were instructed to abstain from any intense or unaccustomed exercise (e.g., training with large eccentric component) and not to use nutritional supplements (e.g., antioxidants) or anti-inflammatory medication for 10 days prior and for the duration of the measurements. The participants prior to testing were fully informed about the purpose, procedures, and possible risks of the study, and written informed consent was obtained. To ensure an adequate population was recruited to reach statistical power (set at 0.8), we calculated effect size (ES) values (Cohen’s *d*) a priori based on previous studies using similar methods [4,6,25,27] for peak vertical ground reaction force (GRF) (ES = 1.05), peak knee joint flexion (ES = 1.05), and peak knee extension moment (ES = 1.46) during landing, total contact phase (ES = 1.09), and jump height (ES = 1.40). The measures with the smallest ES (peak vertical GRF and peak knee flexion, 1.05) were used to calculate sample size. The a priori power calculation using the G power software (Heinrich-Heine-Universität, Düsseldorf, Germany) revealed that the initial sample size required for statistical power 0.8 was 12. All procedures were conducted according to the Declaration of Helsinki and were approved by the University of Thessaly ethics committee (approval code: 1524).

### 2.2. Experimental Design

Participants visited the biomechanics laboratory on 5 different occasions at the same time of the day. In the first (−72 h) and second (−24 h) visits pre-EIMD protocol, baseline measurements were carried out and experimental protocols were performed in visits 3 (0 h; EIMD), 4 (+24 h), and 5 (+48 h) post-EIMD protocol (please refer to Table 1 below). During the first visit, anthropometric measurements were recorded, and the participants were familiarized with the DVJ task and the isokinetic dynamometer. Baseline isometric peak torque was also collected. In the second visit, baseline DVJ kinematics and kinetics were captured. Then, the EIMD protocol was conducted during the third visit (0 h). Given that EIMD indicators increase mainly 24–72 h post-exercise [12,13], follow-up DVJ kinematic and kinetic data were collected 48 h after the EIMD protocol (fifth visit). EIMD indicators included isometric peak torque, muscle soreness, and serum CK activity. Muscle soreness was assessed during the second visit (−24 h), pre-exercise during the third (0 h) visit, and post-exercise during the fourth (+24 h) and fifth (+48 h) visits. Peak torque was measured during the first visit (−72 h), pre-exercise during and immediately after the third visit (0 h), and post-exercise in the fourth (+24 h) and fifth (+48 h) visits. CK activity was evaluated pre-exercise in the third visit (0 h) and post-exercise in the fifth visit (+48 h). The time course for assessing the EIMD indicators was similar to previous research [18,19]. The baseline measurements recorded in the first and second visit were used to conduct a test–retest reliability assessment of the EIMD indicators (isometric peak torque, muscle soreness) based on the measurements collected in the third visit [18,28].

### 2.3. EIMD Protocol

A standardized warm-up included 5 min of jogging on a treadmill (Technogym, Italy) at 2.5 m/s, followed by stretching of the major muscle groups of the lower limbs. A single-joint model was used to induce muscle damage based on a maximal eccentric exercise protocol performed on an isokinetic dynamometer (Cybex-Norm, Ronkonkoma, NY). The isokinetic exercise was conducted in a seated position (100° hip angle) and the trunk, waist, and thighs were stabilized using straps. The knee axis of rotation was aligned with the axis of rotation of the dynamometer’s attachment arm and gravitational corrections were employed. The knee ROM was set between 10 and 100° (0° = full knee extension) to minimize knee discomfort. The EIMD protocol consisted of 5 sets of 15 maximal eccentric voluntary contractions of the knee extensors at 60°/s [29]. A recovery period of 10–12 s was provided between contractions, during which the arm of the isokinetic dynamometer was passively returned to the starting position at 10°. Each set was separated by 3 min of recovery. Eccentric exercise was conducted on one side of the body and then repeated on the contralateral leg after a 10 min of recovery. Almost half of the participants (*n* = 8) exercised their dominant leg first. The dominant leg was determined by asking the participant’s preferred leg when kicking a ball [30]. The selected recovery periods aimed to minimize the effects of acute fatigue and were based on related EIMD research that used similar recovery periods between contractions [31,32], sets [33], and sides [33]. The muscular work for each set of 15 contractions was calculated automatically by the software of the dynamometer. The cumulative work generated over the 5 sets was determined as the sum of the values of each set to ensure that similar muscle damage was induced in both limbs. Visual and verbal feedback of their performance was provided, and participants were instructed to perform as forcefully as possible through the whole ROM.

### 2.4. Indicators of Muscle Damage

Identical procedures were followed during the eccentric exercise to measure isometric torque. Participants performed 2 knee extensor 5-s isometric maximal voluntary contractions of both legs at 70° of knee flexion [24] with 2 min recovery period [33]. The peak torque was determined by obtaining the average of the two 5-s contractions. DOMS was tested for both legs and the participants were asked to mark their perceived soreness level on a scale from 1 (no soreness) to 10 (extremely sore) [29,34], while the same investigator throughout the time course of the measurements palpated the muscle belly and distal region of the quadriceps in a relaxed state. DOMS was calculated as the average muscle soreness of the 3 superficial heads of the quadriceps muscle. CK was assessed using previously reported procedures [29].

### 2.5. Data Collection and DVJ Task

Each participant wore the same pair of shoes during the 2 DVJ testing sessions (24 h pre- and 48 h post-eccentric exercise) to avoid potential influence of footwear. Kinematic data were recorded using a 10-camera 3D motion analysis system (Vicon T-series, Oxford, UK) at 200 Hz. Two force platforms (Bertec 4060-10, OH) captured GRFs at 1000 Hz for the right and left leg synchronized with the kinematic data. Twenty reflective markers were placed bilaterally on the pelvis and lower extremities according to the marker set described in the literature [33]. The symmetrical center of rotation estimation (SCoRE) [35] and the symmetrical axes of rotation approach (SARA) [36] were used to calculate the hip joint center and the optimized knee joint flexion axis, respectively. Following the standardized warm-up, the DVJ task was performed as previously reported [30,37]. The participants landed from a 30-cm platform with both feet contacting the ground simultaneously and immediately performed 3 DVJs with maximum effort. They were allowed to perform 3 practice repetitions prior to data collection. To restrict arm movement, participants were required to place their hands on the chest and a 1 min of recovery between trials was given to avoid acute fatigue. The average of 3 successful trials per participant was used for further data analysis [37].

### 2.6. Data Analyses

Initial contact and take off events during DVJ were defined when the raw vertical GRF was over or below the threshold of 10 N, respectively. Marker trajectories and GRFs were then filtered using a fourth-order low-pass Butterworth filter with a cut-off frequency of 15 Hz [38]. GRF and kinematic data were combined using inverse dynamics to calculate internal moments and powers of the lower limb joints. GRFs were expressed as a percentage of body weight and kinetic data were amplitude-normalized to body mass. The contact phase was divided into 2 phases: (a) the landing phase, defined as the time interval from initial ground contact to peak knee joint flexion angle, and (b) the propulsion phase, defined as the time interval between peak knee flexion and take-off [39]. The variables of interest for this study included sagittal plane lower body joint angles, internal joint moments and powers, as well as vertical GRF and knee joint stiffness (see Table 3 of the Results section for their full description). Knee joint stiffness was defined as the ratio of the knee joint moment change to the knee joint ROM between initial contact and the instant of the peak knee joint angle [39]. Total contact, landing, and propulsion phase durations and jump height performance were also assessed. The latter was computed as the vertical displacement of the pelvis (average of four pelvic markers) between the static standing position and the peak height achieved during the DVJ [40].

### 2.7. Statistical Analyses

Normal distribution was assessed using a Shapiro–Wilk test and found no significant difference, indicating that all datasets were normally distributed. Paired sample *t*-tests between the dominant and non-dominant leg revealed no significant differences, and thus the data of the dominant leg were used for analysis. Coefficient of variation (CV), standard error of measurement (SEm), and intraclass correlation coefficient (R) were used to assess the test–retest reliability of the EIMD indicators (isometric peak torque, muscle soreness) measured on 2 separate occasions prior to eccentric exercise. One-way analyses of variance (ANOVAs) with repeated measures were used to analyze muscle soreness (4 measurements) and isometric peak torque (3 measurements) during different time-points. Paired sample *t*-tests were employed to compare sagittal plane kinematics and kinetics as well as CK activity between pre- and post-eccentric exercise conditions. The significance level was set at *p* < 0.05.

## 3. Results

### 3.1. Muscle Damage Indicators

The CV for isometric peak torque and muscle soreness prior to eccentric exercise were 4.27 and 0%, respectively. The SEm value for isometric peak torque was ± 2.01 Nm and for muscle soreness 0, and the R values were 0.97 and 1.00, respectively. The muscle damage indicators are shown in Table 2. Following the EIMD protocol, a significant decrease was found in isometric peak torque, whilst a significant increase was identified in CK activity and DOMS at all time points post-exercise (*p* < 0.05).

### 3.2. DVJ Biomechanical Measures

Pelvic kinematics and dominant lower limb’s kinematic and kinetic results are demonstrated in Table 3. Regarding the landing phase of the DVJ task, the results revealed significant increase for the anterior pelvic tilt, peak hip, and knee flexion angles pre- (19.74 ± 5.9°; 61.18 ± 10.5°; 84.15 ± 6.2°) and 48 h post- (25.16 ± 6.9°; 69.38 ± 10.3°; 88.66 ± 6.9°) exercise, respectively (*p* < 0.05). On the other hand, no kinematic differences were found for pelvis, hip, and knee joints at initial contact post-exercise (*p* > 0.05). The more flexed knee joint angle observed at the end of the landing phase of the DVJ task was accompanied by decreases in peak vertical GRF (12.3%), knee joint stiffness (28.6%), and peak knee joint internal extension moment (16.2%) and power absorption (17.4%). No differences during this phase were noted in comparison to pre-exercise values for ankle joint kinematics and kinetics (*p* > 0.05). Notable results regarding the propulsion phase of the DVJ task included significant decrements for peak vertical GRF (16.4%) as well as for peak knee (18.9%) and ankle (21.63%) joint power generation (*p* < 0.05), while no alterations were observed for maximum hip joint power generation (*p* > 0.05). Finally, total contact (21.9%), landing (28.9%), and propulsion (16.6%) phase duration and jump height performance (3.7%) during the DVJ task were reduced significantly 48 h post-exercise (*p* < 0.05).

## 4. Discussion

The primary aim of the present study was to investigate the effects of an eccentric localized muscle damage protocol on pelvic and lower extremity sagittal plane biomechanics during a DVJ task. There is a growing interest for strength and conditioning practitioners since many athletes continue to train and participate in athletic events, despite experiencing significant EIMD symptoms. The participants displayed changes in several kinematic and kinetic parameters examined 48 h post-eccentric exercise and therefore the hypothesis that landing mechanics are affected whilst in a muscle damaged state is accepted. Moreover, the movement alterations observed led to a small but significant decrement in the maximum DVJ height performance.

A critical advantage of employing a single-joint model allows the biomechanical changes during landing to be examined as a consequence of localized damage of a specific muscle group. The observed changes in muscle damage indicators (Table 2) confirmed that muscle damage occurred 48 h following the EIMD protocol [13]. These findings are in agreement with those previously reported when similar isokinetic eccentric knee extensor protocols were applied to induce EIMD [28,29]. Furthermore, a bilateral DVJ task was adopted in the present study given that it is consistently used by strength and conditioning practitioners in plyometric training involving the SSC and it is commonly employed in clinical settings to assess and screen injury risk during landing [10,37]. Pre-eccentric exercise DVJ kinematics and kinetics are consistent with those observed in the existing literature [4,27].

During the landing phase, the joints of the lower extremities function to control the momentum of the body’s CoM through joint flexion. Although the impulse of the GRF acts to change the momentum of the CoM during the absorption phase of landing, the dynamics of the actual GRF (peak value and loading rate) can be manipulated by the athlete. The landing pattern can be defined as soft or stiff when the peak knee flexion angle during landing phase is greater than or less than 90°, respectively [41]. Based on this criterion, it was found that participants in the current study employed a slightly stiff landing pattern both before (84.16 ± 6.4°) and after (88.66 ± 7.2°) the muscle damage protocol. However, the knee joint was statistically more flexed 48 h post-exercise, indicating the participants’ preference for a softer landing technique after eccentric EIMD (Table 3). A flexed landing posture improves the ability of the lower extremities to attenuate landing forces. Indeed, the abovementioned peak knee flexion modification produced a substantial reduction in the peak vertical GRF and a concomitant decrease in peak knee joint internal extension moment during the landing phase (Table 3). Based on of the existing literature, a reduced vertical GRF and knee moment while increasing knee angular displacement may reduce the risk of ACL injuries during landing [7,10,42]. It has been suggested that a more erect landing position is associated with an increased anterior tibial shear force induced by the knee extensor muscles through changes in the patellar tendon insertion angle [3]. Limited sagittal plane movement has also been linked to increased frontal plane loading, which is considered a significant risk factor for ACL injury [37]. Thus, greater knee flexion during the absorption phase of landing can reduce anterior tibial shear force; restrict large knee abduction/internal rotation angles and moments; and, as a result, minimize the stress placed on the ACL [3,37].

The data of the present study did not determine whether the increase in peak knee flexion angle is an intentional or unintentional strategy to prevent further musculoskeletal injury or inability to maintain the pre-exercise landing pattern due to EIMD-induced decrement in force-generating capacity (Table 2). Previous research suggested that athletes use a more flexed landing position during running in instances where it is difficult to control the lower limb (i.e., surface irregularities and fatigue) [43,44]. This strategy provides the runners a larger margin for dealing with kinematic errors, but it has an associated metabolic cost that potentially reduces performance [44]. Thus, the observed increased knee joint flexion during landing indicates that EIMD may be another condition that can lead to a similar strategy, in an attempt to increase safety [44]. Moreover, given that the optimum length of tension development in muscle can be shifted to longer muscle lengths following EIMD [45], the increased peak knee flexion angle 48 h post-exercise may be a compensating strategy to achieve optimal muscle length for torque production. However, a greater flexion of the lower joints during absorption may increase the risk of overuse injuries by increasing the energy absorption of the musculotendinous units that are in a lengthened position [46].

Other factors that can possibly explain the increased knee flexion during landing are associated with muscle soreness, altered pre-activity, stiffness modification, and decreased reflex sensitivity. Exhausting sub-maximal rebound series performed on a sledge apparatus have been shown to induce muscle damage and disturb stretch reflex sensitivity, stiffness regulation, and pre-activity of knee muscles for at least 4 days post-exercise [26,27]. The mechanisms related to stretch reflex sensitivity while being in muscle damage state may involve presynaptic inhibition and a reduction in muscle spindle sensitivity [27]. These mechanisms can possibly explain the decreased knee joint stiffness found in the present study (Table 3). A possible explanation with regards to the involvement of DOMS in the observed knee joint flexion post-exercise is that as muscle soreness progresses 48 h after the EIMD protocol, the initial high impact load during landing cannot be tolerated resulting in an increased knee angular displacement. A similar pattern of unloading the knee extensor muscles has been observed during walking following experimentally induced pain [47].

Furthermore, peak anterior pelvic tilt and peak hip flexion angle during landing phase were significantly increased following the EIMD protocol (Table 3). Interestingly, examined hip joint kinetic variables were not altered both during the landing and propulsion phase of the DVJ. In contrast, significant decrements were noted for the respective knee joint kinetic variables (Table 3). Previous research reported the important role of the hip in proximal control of the knee during closed kinetic chain movements [48,49]. It has been documented that greater knee and hip joint flexion and the coupling of the two joints during landing produces softer landings and shields the ACL from excessive loading [50]. Additionally, athletes who employ greater knee and hip flexion exhibit decreased knee valgus angles and knee adductor moments compared to those with limited sagittal plane motion during landing [50]. As a result, the observed increased hip flexion together with the fact that hip extension moment and hip power absorption were not altered following eccentric exercise may represent a strategy to reduce eccentric work of the knee extensors during landing in an attempt to avoid knee collapse, given the eccentric exercise-induced knee extensor strength deficiency. Further, an increased trunk flexion and anterior pelvic tilt were reported to aid the deceleration of the CoM, since trunk flexion is associated with the reduction of impact forces observed during landing [51]. Therefore, increasing anterior pelvic tilt during landing could be a strategy to reduce knee joint loading by transferring the demands on the hip joint, while concomitant trunk flexion and pelvic tilt moves the CoM anteriorly and alters knee joint moment arms [51].

Despite a significant decrease (19.5%) in the force-generating capacity (Table 1) 48 h post eccentric-exercise, jump height performance was not affected to the same extent (3.7% decline; Table 3). The observed reduction in jumping performance was slightly smaller compared to the 4.5–14% declines reported 2–4 days following exhaustive maximal/submaximal SSC exercise protocols performed on a sledge apparatus [26,27,52]. The small decline observed in jumping performance in the present study can in part be explained by the increased contribution of the hip extensor muscles. As described above, hip power absorption during landing phase and hip power generation during propulsion phase remained unaltered, with the latter showing an insignificant trend to increase 48 h post-exercise (Table 3). On the other hand, knee joint power significantly decreased during both landing and propulsion phases. A significant decrease was noted as well for positive ankle joint power during the propulsion phase (Table 3). It is thus likely that during the execution of the DVJ, the participants attempted (intentionally or unintentionally) to minimize the reduction in jumping performance by increasing the contribution of the hip extensor muscles. A similar strategy has been described following SSC-induced muscle damage performed on a sledge apparatus [52]. It should be pointed out, however, that even a small decrement in jumping performance was accompanied by a 18.9% increase in total contact phase duration (Table 3), which may not be conducive to the rapid movements required in many sports.

A limitation of the present study is that the frontal plane was not examined, although specific biomechanical variables in this plane have been associated with non-contact ACL injuries [2,37]. Moreover, inter-individual variability due to differences in EIMD susceptibility should be taken into account, considering that its effect could be augmented when a small sample size is used [53]. Given that external physical load can be perceived differently by each athlete during deceleration when landing from a jump in unusual circumstances such as acute and residual fatigue [9], no systematic change may be found due to inter-individual variability in specific measured outcomes, leading to false conclusions regarding the impact of EIMD on landing biomechanics. Lastly, the sample in the present study is not representative of the whole population, and therefore further research incorporating high-level professional athletes as well as individuals who have performed ACL reconstruction may be also warranted.

## 5. Conclusions

The participants displayed a softer landing pattern during DVJ and a small decrease in jump height performance in response to the applied isokinetic eccentric EIMD protocol. The present study provides new insights on how the landing kinematics and kinetics change during DVJ to compensate the impaired function of the knee extensors following EIMD and residual muscle soreness 48 h post-exercise. The biomechanical outcomes discussed are laboratory-based and thus do not directly translate to real-world environments. However, a practical application for strength and conditioning practitioners pertaining to the design of the training microcycle is that landing technique could be altered (i.e., prolonged contact in conjunction with larger peak knee and hip joint flexion) while performing DVJs (plyometrics) in the presence of muscle damage without notable decline in jumping performance. Future research should examine kinematic, kinetic, and EMG data incorporated in forward dynamic simulations to elucidate the underlying mechanisms behind the observations made in the present study.

## Figures and Tables

**Table 1 sports-09-00024-t001:** Experimental design timeline.

	−72 h	−24 h	0 h	+24 h	+48 h
Eccentric exercise			X		
DVJ biomechanics		X			X
Isometric peak torque	X		X *	X	X
DOMS		X	X	X	X
CK activity			X		X

* Isometric peak torque was examined pre- and post-exercise.

**Table 2 sports-09-00024-t002:** Muscle damage indicators following the exercise-induced muscle damage (EIMD) protocol. Data are reported as mean ± SD.

Muscle Damage Indicators	0 h	Immediately After	+24 h	+48 h
IPT extensors (%)	100 ± 0	75.4 ± 7.0 *	79.7 ± 10.6 *	80.5 ± 9.9 *
CK (U/L)	202 ± 125	NM	NM	2040 ± 1306 *
DOMS extensors	0 ± 0	NM	5.5 ± 1.3 *	6.7 ± 1.6 *

* Significantly different compared to the pre-exercise values (*p* < 0.05). NM: not measured; IPT (%): isometric peak torque expressed as percentage of pre-exercise values.

**Table 3 sports-09-00024-t003:** Biomechanical measures 0 h and 48 h following EIMD protocol (mean ± SD).

	0 h	+48 h	*p*
Peak vertical GRF during landing phase (% BW)	2.04 ± 0.37	1.79 ± 0.33 *	<0.05
Peak vertical GRF during propulsion phase (% BW)	1.89 ± 0.32	1.58 ± 0.25 *	<0.05
Pelvic anterior tilt at initial contact (°)	17.31 ± 4.4	20.19 ± 7.8	NS
Peak anterior pelvic tilt during landing phase (°)	19.74 ± 5.9	25.16 ± 6.9 *	<0.05
Hip flexion angle at initial contact (°)	43.39 ± 6.4	44.49 ± 7.5	NS
Peak hip flexion angle during landing phase (°)	61.18 ± 10.5	69.38 ± 10.3 *	<0.05
Peak hip extension moment during landing phase (Nm/kg)	2.46 ± 0.69	2.41 ± 0.51	NS
Peak hip power absorption during landing phase (W/kg)	−10.8 ± 5.5	−10.4 ± 4.2	NS
Peak hip power generation during propulsion phase (W/kg)	5.6 ± 1.8	6.4 ± 1.7	NS
Knee flexion angle at initial contact (°)	41.61 ± 5.9	40.37 ± 7.4	NS
Peak knee flexion angle during landing phase (°)	84.15 ± 6.2	88.66 ± 6.9 *	<0.05
Peak knee extension moment during landing phase (Nm/kg)	3.02 ± 0.44	2.53 ± 0.53 *	<0.05
Peak knee power absorption during landing phase (W/kg)	−20.65 ± 3.9	−17.06 ± 4.2 *	<0.05
Peak knee power generation during propulsion phase (W/kg)	18.19 ± 2.9	14.76 ± 3.9 *	<0.05
Ankle angle at initial contact (°)	−4.07 ± 8.6	−9.02 ± 6.7	NS
Peak ankle angle during landing phase (°)	37.69 ± 4.2	35.51 ± 3.1	NS
Peak ankle moment during landing phase (Nm/kg)	2.20 ± 0.64	1.97 ± 0.35	NS
Peak ankle power absorption during landing phase (W/kg)	−12.93 ± 4.3	−11.66 ± 3.2	NS
Peak ankle power generation during propulsion phase (W/kg)	17.38 ± 3.0	13.62 ± 2.7 *	<0.05
Knee joint stiffness (Nm/kg∙deg)	0.077 ± 0.03	0.055 ± 0.02 *	<0.05
Total contact phase duration (ms)	313.2 ± 60.7	381.7 ± 76.6 *	<0.05
Landing phase duration (ms)	134.4 ± 35.4	173.3 ± 44.1 *	<0.05
Propulsion phase duration (ms)	178.8 ± 28.1	208.4 ± 35.7 *	<0.05
Jump height (cm)	33.38 ± 1.21	32.15 ± 1.4 *	<0.05

BW, body weight; NS, non-significant. Ankle dorsiflexion and joint power generation are represented as positive and joint power absorption is represented as negative. * Significantly different compared to the pre-exercise values (*p* < 0.05).

## Data Availability

The data supporting reported results can be found in https://zenodo.org/record/4397664#.X-qfRen7RhE (accessed on 3 February 2021).

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
