# Peer review of "Altered Drop Jump Landing Biomechanics Following Eccentric Exercise-Induced Muscle Damage"

_sports, 2021, doi:10.3390/sports9020024_

Round 1

Reviewer 1 Report

Congratulations on this research. In this work, the importance of eccentric muscle damage in landing is studied, which is very important in some sports, so the subject is very interesting. However, some changes need to be made to improve it:

Materials and Methods:

2.2. Experimental Design

The experimental design time line must be more explained. For example, why were some measurements + 24h and + 48h made and not all + 48h?

2.3. EIMD Protocol

Does the recovery period between contractions and series belong to Deli et al. protocol? If yes, please include the reference. If not, briefly explain why that duration

How was the accumulated muscle work generated in the 5 sets calculated?

2.4. Indicators of Muscle Damage

The same as in 2.3. EIMD protocol about the rest duration reference

Table 2:

IPT extensors (%) is expressed as % of pre-exercise values (line 217). However, the title of the table says that the data are reported as mean ± SD (line 215). It must be better explained

Discussion:

Add as a limitation of the study that the sample is not representative of the whole, so the results should be taken with caution.

As I have discussed previously, this is an interesting study that may allow for development and research on motor control and injury prevention.

Thanks for this interesting contribution.

Reviewer 2 Report

First of all I would like to commend the authors by the innovative work and the clinical approach they give to the study. But, from my point of view there are:

I wonder how to determine sample size?
I recommend that you provide contents (ex. Population size, confidence interval, confidence level, standard of deviation, etc.) of sample size calculation into a text.

I recommend that you determine the SEM and CV should be provided for the reliability assessments. ICC alone is not sufficient (e.g., Hopkins, Sports Med, 2000).

Reviewer 3 Report

The paper presents a reserach study that investigated the effects of knee localized muscle on sagittal plane landing biomechanics during drop vertical jump (DVJ).

The paper is well written and clearly structured. The method is well expleained and the data analysis approach is very good for the validation. The discussion is very clear and deep, but the conclusion should be improved and extended.

The paper can be accepted after a minor review.

Reviewer 4 Report

I regret to inform you that I can’t recommend this submission for publication considering several flaws. Particularly:

1) The study design does not meet the rationale and aim of the paper (the introduction and discussion present data on athletes and sports, whereas the sample size is consisted by physically active men) resulting in lack of external validity of the findings.

2) The authors should present a detailed power analysis estimating the sample size.

3) Although the association with injuries is reported in many instances, no data on injuries are shown.

I would encourage the authors to address these concerns before submitting a revised version.

Round 2

Reviewer 1 Report

The proposed changes have been applied and doubts have been clarified, improving the role. Nice job.

Reviewer 4 Report

Despite the effort of the authors to address my concerns, I am not convinced about the external validity of the findings.